# Fixed-Rank Approximation of a
# Positive-Semidefinite Matrix from Streaming Data

**Joel A. Tropp**
Caltech
jtropp@caltech.edu

**Alp Yurtsever**
EPFL
alp.yurtsever@epfl.ch

**Madeleine Udell**
Cornell
mru8@cornell.edu

**Volkan Cevher**
EPFL
volkan.cevher@epfl.ch

## Abstract

Several important applications, such as streaming PCA and semidefinite programming, involve a large-scale positive-semidefinite (psd) matrix that is presented as a sequence of linear updates. Because of storage limitations, it may only be possible to retain a sketch of the psd matrix. This paper develops a new algorithm for fixed-rank psd approximation from a sketch. The approach combines the Nyström approximation with a novel mechanism for rank truncation. Theoretical analysis establishes that the proposed method can achieve any prescribed relative error in the Schatten 1-norm and that it exploits the spectral decay of the input matrix. Computer experiments show that the proposed method dominates alternative techniques for fixed-rank psd matrix approximation across a wide range of examples.

## 1 Motivation

In recent years, researchers have studied many applications where a large positive-semidefinite (psd) matrix is presented as a series of linear updates. A recurring theme is that we only have space to store a small summary of the psd matrix, and we must use this information to construct an accurate psd approximation with specified rank. Here are two important cases where this problem arises.

**Streaming Covariance Estimation.** Suppose that we receive a stream $\boldsymbol{h}_1, \boldsymbol{h}_2, \boldsymbol{h}_3, \dots \in \mathbb{R}^n$ of high-dimensional vectors. The psd sample covariance matrix of these vectors has the linear dynamics

$$\boldsymbol{A}^{(0)} \leftarrow \mathbf{0} \quad \text{and} \quad \boldsymbol{A}^{(i)} \leftarrow (1 - i^{-1})\boldsymbol{A}^{(i-1)} + i^{-1}\boldsymbol{h}_i\boldsymbol{h}_i^*.$$

When the dimension $n$ and the number of vectors are both large, it is not possible to store the vectors or the sample covariance matrix. Instead, we wish to maintain a small summary that allows us to compute the rank-$r$ psd approximation of the sample covariance matrix $\boldsymbol{A}^{(i)}$ at a specified instant $i$. This problem and its variants are often called *streaming PCA* [3, 12, 14, 15, 25, 32].

**Convex Low-Rank Matrix Optimization with Optimal Storage.** A primary application of semidefinite programming (SDP) is to search for a rank-$r$ psd matrix that satisfies additional constraints. Because of storage costs, SDPs are difficult to solve when the matrix variable is large. Recently, Yurtsever et al. [44] exhibited the first provable algorithm, called SketchyCGM, that produces a rank-$r$ approximate solution to an SDP *using optimal storage*.

Implicitly, SketchyCGM forms a sequence of approximate psd solutions to the SDP via the iteration

$$\boldsymbol{A}^{(0)} \leftarrow \mathbf{0} \quad \text{and} \quad \boldsymbol{A}^{(i)} \leftarrow (1 - \eta_i)\boldsymbol{A}^{(i-1)} + \eta_i\boldsymbol{h}_i\boldsymbol{h}_i^*.$$

The step size $\eta_i = 2/(i + 2)$, and the vectors $\boldsymbol{h}_i$ do not depend on the matrices $\boldsymbol{A}^{(i)}$. In fact, SketchyCGM only maintains a small summary of the evolving solution $\boldsymbol{A}^{(i)}$. When the iteration terminates, SketchyCGM computes a rank-$r$ psd approximation of the final iterate using the method described by Tropp et al. [37, Alg. 9].

## 1.1 Notation and Background

The scalar field $\mathbb{F} = \mathbb{R}$ or $\mathbb{F} = \mathbb{C}$. Define $\alpha(\mathbb{R}) = 1$ and $\alpha(\mathbb{C}) = 0$. The asterisk $^*$ is the (conjugate) transpose, and the dagger $^\dagger$ denotes the Moore–Penrose pseudoinverse. The notation $\boldsymbol{A}^{1/2}$ refers to the unique psd square root of a psd matrix $\boldsymbol{A}$. For $p \in [1, \infty]$, the Schatten $p$-norm $\| \cdot \|_p$ returns the $\ell_p$ norm of the singular values of a matrix. As usual, $\sigma_r$ refers to the $r$th largest singular value.

For a nonnegative integer $r$, the phrase "rank-$r$" and its variants mean "rank at most $r$." For a matrix $\boldsymbol{M}$, the symbol $[\![\boldsymbol{M}]\!]_r$ denotes a (simultaneous) best rank-$r$ approximation of the matrix $\boldsymbol{M}$ with respect to any Schatten $p$-norm. We can take $[\![\boldsymbol{M}]\!]_r$ to be any $r$-truncated singular value decomposition (SVD) of $\boldsymbol{M}$ [24, Sec. 6]. Every best rank-$r$ approximation of a psd matrix is psd.

## 2 Sketching and Fixed-Rank PSD Approximation

We begin with a streaming data model for a psd matrix that evolves via a sequence of general linear updates, and it describes a randomized linear sketch for tracking the psd matrix. To compute a fixed-rank psd approximation, we develop an algorithm based on the Nyström method [40], a technique from the literature on kernel methods. In contrast to previous approaches, **our algorithm uses a distinct mechanism to truncate the rank of the approximation.**

**The Streaming Model.** Fix a rank parameter $r$ in the range $1 \leq r \leq n$. Initially, the psd matrix $\boldsymbol{A} \in \mathbb{F}^{n \times n}$ equals a known psd matrix $\boldsymbol{A}_{\text{init}} \in \mathbb{F}^{n \times n}$. Then $\boldsymbol{A}$ evolves via a series of linear updates:

$$\boldsymbol{A} \leftarrow \theta_1 \boldsymbol{A} + \theta_2 \boldsymbol{H} \quad \text{where} \quad \theta_i \in \mathbb{R}, \quad \boldsymbol{H} \in \mathbb{F}^{n \times n} \text{ is (conjugate) symmetric.} \quad (2.1)$$

In many applications, the innovation $\boldsymbol{H}$ is low-rank and/or sparse. We assume that the evolving matrix $\boldsymbol{A}$ always remains psd. At one given instant, we must produce an accurate rank-$r$ approximation of the psd matrix $\boldsymbol{A}$ induced by the stream of linear updates.

**The Sketch.** Fix a sketch size parameter $k$ in the range $r \leq k \leq n$. Independent from $\boldsymbol{A}$, we draw and fix a random test matrix

$$\boldsymbol{\Omega} \in \mathbb{F}^{n \times k}. \quad (2.2)$$

See Sec. 3 for a discussion of possible distributions. The sketch of the matrix $\boldsymbol{A}$ takes the form

$$\boldsymbol{Y} = \boldsymbol{A}\boldsymbol{\Omega} \in \mathbb{F}^{n \times k}. \quad (2.3)$$

The sketch (2.3) supports updates of the form (2.1):

$$\boldsymbol{Y} \leftarrow \theta_1 \boldsymbol{Y} + \theta_2 \boldsymbol{H}\boldsymbol{\Omega}. \quad (2.4)$$

To find a good rank-$r$ approximation, we must set the sketch size $k$ larger than $r$. But storage costs and computation also increase with $k$. One of our main contributions is to clarify the role of $k$.

Under the model (2.1), it is more or less necessary to use a randomized linear sketch to track $\boldsymbol{A}$ [28]. For psd matrices, sketches of the form (2.2)–(2.3) appear explicitly in Gittens's work [16, 17, 19]. Tropp et al. [37] relies on a more complicated sketch developed in [7, 42].

**The Nyström Approximation.** The Nyström method is a general technique for low-rank psd matrix approximation. Various instantiations appear in the papers [5, 11, 13, 16, 17, 19, 22, 27, 34, 40].

Here is the application to the present situation. Given the test matrix $\boldsymbol{\Omega}$ and the sketch $\boldsymbol{Y} = \boldsymbol{A}\boldsymbol{\Omega}$, the Nyström method constructs a rank-$k$ psd approximation of the psd matrix $\boldsymbol{A}$ via the formula

$$\hat{\boldsymbol{A}}^{\text{nys}} = \boldsymbol{Y}(\boldsymbol{\Omega}^* \boldsymbol{Y})^\dagger \boldsymbol{Y}^*. \quad (2.5)$$

In most work on the Nyström method, the test matrix $\boldsymbol{\Omega}$ depends adaptively on $\boldsymbol{A}$, so these approaches are not valid in the streaming setting. Gittens's framework [16, 17, 19] covers the streaming case.

**Fixed-Rank Nyström Approximation: Prior Art.** To construct a Nyström approximation with exact rank $r$ from a sketch of size $k$, the standard approach is to truncate the center matrix to rank $r$:

$$\hat{\boldsymbol{A}}_r^{\text{nysfix}} = \boldsymbol{Y}([\![\boldsymbol{\Omega}^* \boldsymbol{Y}]\!]_r)^\dagger \boldsymbol{Y}^*. \quad (2.6)$$

The truncated Nyström approximation (2.6) appears in the many papers, including [5, 11, 18, 34]. We have found (Sec. 5) that the truncation method (2.6) performs poorly in the present setting. This observation motivated us to search for more effective techniques.

**Fixed-Rank Nyström Approximation: Proposal.** The purpose of this paper is to develop, analyze, and evaluate a new approach for fixed-rank approximation of a psd matrix under the streaming model. We propose a more intuitive rank-$r$ approximation:

$$\hat{A}_r = [\![\hat{A}^{\mathrm{nys}}]\!]_r. \tag{2.7}$$

That is, we report a best rank-$r$ approximation of the full Nyström approximation (2.5).

This "matrix nearness" approach to fixed-rank approximation appears in the papers [21, 22, 37]. The combination with the Nyström method (2.5) is totally natural. Let us emphasize that the approach (2.7) also applies to Nyström approximations outside the streaming setting.

**Summary of Contributions.** This paper contains a number of advances over the prior art:

1. We propose a new technique (2.7) for truncating the Nyström approximation to rank $r$. This formulation differs from the published literature on fixed-rank Nyström approximations.
2. We present a stable numerical implementation of (2.7) based on the best practices outlined in the paper [27]. This approach is essential for achieving high precision! (Sec. 3)
3. We establish informative error bounds for the method (2.7). In particular, we prove that it attains $(1 + \varepsilon)$-relative error in the Schatten 1-norm when $k = \Theta(r/\varepsilon)$. (Sec. 4)
4. We document numerical experiments on real and synthetic data to demonstrate that our method dominates existing techniques [18, 37] for fixed-rank psd approximation. (Sec. 5)

Psd matrix approximation is a ubiquitous problem, so we expect these results to have a broad impact.

**Related Work.** Randomized algorithms for **low-rank matrix approximation** were proposed in the late 1990s and developed into a technology in the 2000s; see [22, 30, 41]. In the absence of constraints, such as streaming, we recommend the general-purpose methods from [22, 23, 27].

Algorithms for low-rank matrix approximation in the important **streaming data** setting are discussed in [4, 7, 8, 15, 22, 37, 41, 42]. Few of these methods are designed for psd matrices.

**Nyström methods** for low-rank psd matrix approximation appear in [11, 13, 16, 17, 19, 22, 26, 34, 37, 40, 43]. These works mostly concern kernel matrices; they do not focus on the streaming model.

We are only aware of a few papers [16, 17, 19, 37] on algorithms for **psd matrix approximation** that operate under the **streaming model** (2.1). These papers form the comparison group.

After this paper was submitted, we learned about two **contemporary works** [35, 39] that propose the fixed-rank approximation (2.7) in the context of kernel methods. Our research is distinctive because we focus on the streaming setting, we obtain precise error bounds, we address numerical stability, and we include an exhaustive empirical evaluation.

Finally, let us mention two very recent **theoretical papers** [6, 33] that present existential results on algorithms for fixed-rank psd matrix approximation. The approach in [6] is only appropriate for sparse input matrices, while the work [33] is not valid in the streaming setting.

## 3 Implementation

**Distributions for the Test Matrix.** To ensure that the sketch is informative, we must draw the test matrix (2.2) at random from a suitable distribution. The choice of distribution determines the computational requirements for the sketch (2.3), the linear updates (2.4), and the matrix approximation (2.7). It also affects the quality of the approximation (2.7). Let us outline some of the most useful distributions. A full discussion is outside the scope of our work, but see [17, 19, 22, 29, 30, 37, 41].

**Isotropic Models.** Mathematically, the most natural model is to construct a test matrix $\mathbf{\Omega} \in \mathbb{F}^{n \times k}$ whose range is a uniformly random $k$-dimensional subspace in $\mathbb{F}^n$. There are two approaches:

1. **Gaussian.** Draw each entry of the matrix $\mathbf{\Omega} \in \mathbb{F}^{n \times k}$ independently at random from the standard normal distribution on $\mathbb{F}$.
2. **Orthonormal.** Draw a Gaussian matrix $\mathbf{G} \in \mathbb{F}^{n \times k}$, as above. Compute a thin orthogonal–triangular factorization $\mathbf{G} = \mathbf{\Omega R}$ to obtain the test matrix $\mathbf{\Omega} \in \mathbb{F}^{n \times k}$. Discard $\mathbf{R}$.

Gaussian and orthonormal test matrices both require storage of $kn$ floating-point numbers in $\mathbb{F}$ for the test matrix $\mathbf{\Omega}$ and another $kn$ floating-point numbers for the sketch $\mathbf{Y}$. In both cases, the cost of multiplying a vector in $\mathbb{F}^n$ into $\mathbf{\Omega}$ is $\Theta(kn)$ floating-point operations.

---

**Algorithm 1** *Sketch Initialization.* Implements (2.2)–(2.3) with a random orthonormal test matrix.

---

**Input:** Positive-semidefinite input matrix $\boldsymbol{A} \in \mathbb{F}^{n \times n}$; sketch size parameter $k$
**Output:** Constructs test matrix $\boldsymbol{\Omega} \in \mathbb{F}^{n \times k}$ and sketch $\boldsymbol{Y} = \boldsymbol{A}\boldsymbol{\Omega} \in \mathbb{F}^{n \times k}$

1   **local:** $\boldsymbol{\Omega}, \boldsymbol{Y}$                                                   ▷ Internal variables for NYSTROMSKETCH
2   **function** NYSTROMSKETCH($\boldsymbol{A}; k$)                                                     ▷ Constructor
3       **if** $\mathbb{F} = \mathbb{R}$ **then**
4           $\boldsymbol{\Omega} \leftarrow \texttt{randn}(n, k)$
5       **if** $\mathbb{F} = \mathbb{C}$ **then**
6           $\boldsymbol{\Omega} \leftarrow \texttt{randn}(n, k) + \texttt{i} * \texttt{randn}(n, k)$
7       $\boldsymbol{\Omega} \leftarrow \texttt{orth}(\boldsymbol{\Omega})$                              ▷ Improve numerical stability
8       $\boldsymbol{Y} \leftarrow \boldsymbol{A}\boldsymbol{\Omega}$

---

**Algorithm 2** *Linear Update.* Implements (2.4).

---

**Input:** Scalars $\theta_1, \theta_2 \in \mathbb{R}$ and conjugate symmetric $\boldsymbol{H} \in \mathbb{F}^{n \times n}$
**Output:** Updates sketch to reflect linear innovation $\boldsymbol{A} \leftarrow \theta_1 \boldsymbol{A} + \theta_2 \boldsymbol{H}$

1   **local:** $\boldsymbol{\Omega}, \boldsymbol{Y}$                                                   ▷ Internal variables for NYSTROMSKETCH
2   **function** LINEARUPDATE($\theta_1, \theta_2, \boldsymbol{H}$)
3       $\boldsymbol{Y} \leftarrow \theta_1 \boldsymbol{Y} + \theta_2 \boldsymbol{H}\boldsymbol{\Omega}$

---

For isotropic models, we can analyze the approximation (2.7) in detail. In exact arithmetic, Gaussian and isotropic test matrices yield identical Nyström approximations (Supplement). In floating-point arithmetic, orthonormal matrices are more stable for large $k$, but we can generate Gaussian matrices with less arithmetic and communication. References for isotropic test matrices include [21, 22, 31].

**Subsampled Scrambled Fourier Transform (SSFT).** One shortcoming of the isotropic models is the cost of storing the test matrix and the cost of multiplying a vector into the test matrix. We can often reduce these costs using an SSFT test matrix. An SSFT takes the form

$$\boldsymbol{\Omega} = \boldsymbol{\Pi}_1 \boldsymbol{F} \boldsymbol{\Pi}_2 \boldsymbol{F} \boldsymbol{R} \in \mathbb{F}^{n \times k}. \tag{3.1}$$

The $\boldsymbol{\Pi}_i \in \mathbb{F}^{n \times n}$ are independent, signed permutation matrices,[1] chosen uniformly at random. The matrix $\boldsymbol{F} \in \mathbb{F}^{n \times n}$ is a discrete Fourier transform ($\mathbb{F} = \mathbb{C}$) or a discrete cosine transform ($\mathbb{F} = \mathbb{R}$). The matrix $\boldsymbol{R} \in \mathbb{F}^{n \times k}$ is a restriction to $k$ coordinates, chosen uniformly at random.

An SSFT $\boldsymbol{\Omega}$ requires only $\Theta(n)$ storage, but the sketch $\boldsymbol{Y}$ still requires storage of $kn$ numbers. We can multiply a vector in $\mathbb{F}^n$ into $\boldsymbol{\Omega}$ using $\Theta(n \log n)$ arithmetic operations via an FFT or FCT algorithm. Thus, for most choices of sketch size $k$, the SSFT improves over the isotropic models.

In practice, the SSFT yields matrix approximations whose quality is identical to those we obtain with an isotropic test matrix (Sec. 5). Although the analysis for SSFTs is less complete, the empirical evidence confirms that the theory for isotropic models also offers excellent guidance for SSFTs. References for SSFTs and related test matrices include [1, 2, 9, 22, 29, 36, 42].

**Numerically Stable Implementation.** It requires care to compute the fixed-rank approximation (2.7). The supplement shows that a poor implementation may produce an approximation with 100% error!

Let us outline a numerically stable and very accurate implementation of (2.7), based on an idea from [27, 38]. Fix a small parameter $\nu > 0$. Instead of approximating the psd matrix $\boldsymbol{A}$ directly, we approximate the shifted matrix $\boldsymbol{A}_\nu = \boldsymbol{A} + \nu\boldsymbol{I}$ and then remove the shift. Here are the steps:

1. Construct the shifted sketch $\boldsymbol{Y}_\nu = \boldsymbol{Y} + \nu\boldsymbol{\Omega}$.
2. Form the matrix $\boldsymbol{B} = \boldsymbol{\Omega}^* \boldsymbol{Y}_\nu$.
3. Compute a Cholesky decomposition $\boldsymbol{B} = \boldsymbol{C}\boldsymbol{C}^*$.
4. Compute $\boldsymbol{E} = \boldsymbol{Y}_\nu \boldsymbol{C}^{-1}$ by back-substitution.
5. Compute the (thin) singular value decomposition $\boldsymbol{E} = \boldsymbol{U}\boldsymbol{\Sigma}\boldsymbol{V}^*$.
6. Form $\hat{\boldsymbol{A}}_r = \boldsymbol{U}[\![\boldsymbol{\Sigma}^2 - \nu\boldsymbol{I}]\!]_r \boldsymbol{U}^*$.

**Algorithm 3** *Fixed-Rank PSD Approximation.* Implements (2.7).

---

**Input:** Matrix $A$ in sketch must be psd; rank parameter $1 \leq r \leq k$
**Output:** Returns factors $U \in \mathbb{F}^{n \times r}$ with orthonormal columns and nonnegative, diagonal $\Lambda \in \mathbb{F}^{r \times r}$
    that form a rank-$r$ psd approximation $\hat{A}_r = U\Lambda U^*$ of the sketched matrix $A$

1   **local:** $\Omega, Y$                                                  ▷ Internal variables for NYSTROMSKETCH
2   **function** FIXEDRANKPSDAPPROX($r$)
3       $\nu \leftarrow \mu \, \texttt{norm}(Y)$                              ▷ $\mu = 2.2 \cdot 10^{-16}$ in double precision
4       $Y \leftarrow Y + \nu\Omega$                         ▷ Sketch of shifted matrix $A + \nu\mathbf{I}$
5       $B \leftarrow \Omega^* Y$
6       $C \leftarrow \texttt{chol}((B + B^*)/2)$                   ▷ Force symmetry
7       $(U, \Sigma, \sim) \leftarrow \texttt{svd}(Y/C, \texttt{'econ'})$    ▷ Solve least squares problem; form thin SVD
8       $U \leftarrow U(:, \ \texttt{1:r})$ and $\Sigma \leftarrow \Sigma(\texttt{1:r}, \ \texttt{1:r})$         ▷ Truncate to rank $r$
9       $\Lambda \leftarrow \max\{0, \Sigma^2 - \nu\mathbf{I}\}$        ▷ Square to get eigenvalues; remove shift
10   **return** $(U, \Lambda)$

---

The pseudocode addresses some additional implementation details. Related, but distinct, methods were proposed by Williams & Seeger [40] and analyzed in Gittens's thesis [17].

**Pseudocode.** We present detailed pseudocode for the sketch (2.2)–(2.4) and the implementation of the fixed-rank psd approximation (2.7) described above. For simplicity, we only elaborate the case of a random orthonormal test matrix; we have also developed an SSFT implementation for empirical testing. The pseudocode uses both mathematical notation and MATLAB 2017A functions.

**Algorithms and Computational Costs.** Algorithm 1 constructs a random orthonormal test matrix, and computes the sketch (2.3) of an input matrix. The test matrix and sketch require the storage of $2kn$ floating-point numbers. Owing to the orthogonalization step, the construction of the test matrix requires $\Theta(k^2 n)$ floating-point operations. For a general input matrix, the sketch requires $\Theta(kn^2)$ floating-point operations; this cost can be removed by initializing the input matrix to zero.

Algorithm 2 implements the linear update (2.4) to the sketch. Nominally, the computation requires $\Theta(kn^2)$ arithmetic operations, but this cost can be reduced when $H$ has structure (e.g., low rank). Using the SSFT test matrix (3.1) also reduces this cost.

Algorithm 3 computes the rank-$r$ psd approximation (2.7). This method requires additional storage of $\Theta(kn)$. The arithmetic cost is $\Theta(k^2 n)$ operations, which is dominated by the SVD of the matrix $E$.

## 4   Theoretical Results

**Relative Error Bound.** Our first result is an accurate bound for the expected Schatten 1-norm error in the fixed-rank psd approximation (2.7).

**Theorem 4.1** (Fixed-Rank Nyström: Relative Error). *Assume $1 \leq r < k \leq n$. Let $A \in \mathbb{F}^{n \times n}$ be a psd matrix. Draw a test matrix $\Omega \in \mathbb{F}^{n \times k}$ from the Gaussian or orthonormal distribution, and form the sketch $Y = A\Omega$. Then the approximation $\hat{A}_r$ given by (2.5) and (2.7) satisfies*

$$\mathbb{E}\,\|A - \hat{A}_r\|_1 \leq \left(1 + \frac{r}{k - r - \alpha}\right) \cdot \|A - [\![A]\!]_r\|_1; \tag{4.1}$$

$$\mathbb{E}\,\|A - \hat{A}_r\|_\infty \leq \|A - [\![A]\!]_r\|_\infty + \frac{r}{k - r - \alpha} \cdot \|A - [\![A]\!]_r\|_1. \tag{4.2}$$

*The quantities $\alpha(\mathbb{R}) = 1$ and $\alpha(\mathbb{C}) = 0$. Similar results hold with high probability.*

The proof appears in the supplement.

In contrast to all previous analyses of randomized Nyström methods, Theorem 4.1 yields explicit, sharp constants. (The contemporary work [39, Thm. 1] contains only a less precise variant of (4.1).) As a consequence, the formulae (4.1)–(4.2) offer an *a priori* mechanism for selecting the sketch size $k$ to achieve a desired error bound. In particular, for each $\varepsilon > 0$,

$$k = (1 + \varepsilon^{-1})r + \alpha \quad \text{implies} \quad \mathbb{E}\,\|A - \hat{A}_r\|_1 \leq (1 + \varepsilon) \cdot \|A - [\![A]\!]_r\|_1.$$

Thus, we can attain an arbitrarily small relative error in the Schatten 1-norm. In the streaming setting, the scaling $k = \Theta(r/\varepsilon)$ is optimal for this result [14, Thm. 4.2]. Furthermore, it is impossible [41, Sec. 6.2] to obtain "pure" relative error bounds in the Schatten $\infty$-norm unless $k = \Omega(n)$.

**The Role of Spectral Decay.** To circumvent these limitations, it is necessary to develop a different kind of error bound. Our second result shows that the fixed-rank psd approximation (2.7) automatically exploits decay in the spectrum of the input matrix.

**Theorem 4.2** (Fixed-Rank Nyström: Spectral Decay)**.** *Instate the notation and assumptions of Theorem 4.1. Then*

$$\mathbb{E}\,\|\boldsymbol{A} - \hat{\boldsymbol{A}}_r\|_1 \leq \|\boldsymbol{A} - [\![\boldsymbol{A}]\!]_r\|_1 + 2\min_{\varrho < k - \alpha}\left[\left(1 + \frac{\varrho}{k - \varrho - \alpha}\right) \cdot \|\boldsymbol{A} - [\![\boldsymbol{A}]\!]_\varrho\|_1\right]; \qquad (4.3)$$

$$\mathbb{E}\,\|\boldsymbol{A} - \hat{\boldsymbol{A}}_r\|_\infty \leq \|\boldsymbol{A} - [\![\boldsymbol{A}]\!]_r\|_\infty + 2\min_{\varrho < k - \alpha}\left[\left(1 + \frac{\varrho}{k - \varrho - \alpha}\right) \cdot \|\boldsymbol{A} - [\![\boldsymbol{A}]\!]_\varrho\|_1\right]. \qquad (4.4)$$

*The index $\varrho$ ranges over the natural numbers.*

The proof of Theorem 4.2 appears in the supplement.

Here is one way to understand this result. As the index $\varrho$ increases, the quantity $\varrho/(k - \varrho - \alpha)$ increases while the rank-$\varrho$ approximation error decreases. Theorem 4.2 states that the approximation (2.7) automatically achieves the best tradeoff between these two terms. When the spectrum of $\boldsymbol{A}$ decays, the rank-$\varrho$ approximation error may be far smaller than the rank-$r$ approximation error. In this case, Theorem 4.2 is tighter than Theorem 4.1, although the prediction is more qualitative.

**Additional Results.** The proofs can be extended to obtain high-probability bounds, as well as results for other Schatten norms or for other test matrices (Supplement).

# 5 Numerical Performance

**Experimental Setup.** In many streaming applications, such as [44], it is essential that the sketch uses as little memory as possible and that the psd approximation achieves the best possible error. For the methods we consider, the arithmetic costs of linear updates and psd approximation are roughly comparable. Therefore, we only assess storage and accuracy.

For the numerical experiments, the field $\mathbb{F} = \mathbb{C}$ except when noted explicitly. Choose a psd input matrix $\boldsymbol{A} \in \mathbb{F}^{n \times n}$ and a target rank $r$. Then fix a sketch size parameter $k$ with $r \leq k \leq n$. For each trial, draw the test matrix $\boldsymbol{\Omega}$ from the orthonormal or the SSFT distribution, and form the sketch $\boldsymbol{Y} = \boldsymbol{A}\boldsymbol{\Omega}$ of the input matrix. Using Algorithm 3, compute the rank-$r$ psd approximation $\hat{\boldsymbol{A}}_r$ defined in (2.7). We evaluate the performance using the relative error metric:

$$\text{Schatten } p\text{-norm relative error} \quad = \quad \frac{\|\boldsymbol{A} - \hat{\boldsymbol{A}}_r\|_p}{\|\boldsymbol{A} - [\![\boldsymbol{A}]\!]_r\|_p} - 1. \qquad (5.1)$$

We perform 20 independent trials and report the average error.

We compare our method (2.7) with the standard truncated Nyström approximation (2.6); the best reference for this type of approach is [18, Sec. 2.2]. The approximation (2.6) is constructed from the same sketch as (2.7), so the experimental procedure is identical.

We also consider the sketching method and psd approximation algorithm [37, Alg. 9] based on earlier work from [7, 22, 42]. We implemented this sketch with orthonormal matrices and also with SSFT matrices. The sketch has two different parameters $(k, \ell)$, so we select the parameters that result in the minimum relative error. Otherwise, the experimental procedure is the same.

We apply the methods to representative input matrices; see the Supplement for plots of the spectra.

**Synthetic Examples.** The synthetic examples are **diagonal** with dimension $n = 10^3$; results for larger and non-diagonal matrices are similar. These matrices are parameterized by an effective rank parameter $R$, which takes values in $\{5, 10, 20\}$. We compute approximations with rank $r = 10$.

    1. **Low-Rank + PSD Noise.** These matrices take the form

$$\boldsymbol{A} = \operatorname{diag}(\underbrace{1, \ldots, 1}_{R}, 0, \ldots, 0) + \xi n^{-1}\boldsymbol{W} \in \mathbb{F}^{n \times n}.$$

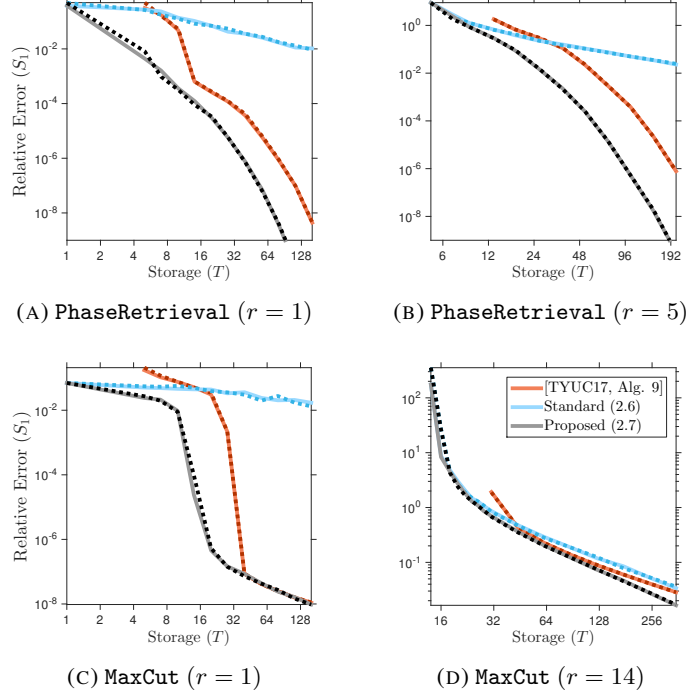

(A) PhaseRetrieval $(r = 1)$       (B) PhaseRetrieval $(r = 5)$

(C) MaxCut $(r = 1)$       (D) MaxCut $(r = 14)$

FIGURE 5.1: **Application Examples, Approximation Rank $r$, Schatten $1$-Norm Error.** The data series show the performance of three algorithms for rank-$r$ psd approximation. **Solid lines** are generated from the Gaussian sketch; **dashed lines** are from the SSFT sketch. Each panel displays the Schatten 1-norm relative error (5.1) as a function of storage cost $T$. See Sec. 5 for details.

The matrix $\boldsymbol{W} \in \mathbb{F}^{n \times n}$ has the WISHART$(n, n; \mathbb{F})$ distribution; that is, $\boldsymbol{W} = \boldsymbol{G}\boldsymbol{G}^*$ where $\boldsymbol{G} \in \mathbb{F}^{n \times n}$ is standard normal. The parameter $\xi$ controls the signal-to-noise ratio. We consider three examples: LowRankLowNoise $(\xi = 10^{-4})$, LowRankMedNoise $(\xi = 10^{-2})$, LowRankHiNoise $(\xi = 10^{-1})$.

2. **Polynomial Decay.** These matrices take the form

$$\boldsymbol{A} = \operatorname{diag}(\underbrace{1, \ldots, 1}_{R}, 2^{-p}, 3^{-p}, \ldots, (n - R + 1)^{-p}) \in \mathbb{F}^{n \times n}.$$

The parameter $p > 0$ controls the rate of polynomial decay. We consider three examples: PolyDecaySlow $(p = 0.5)$, PolyDecayMed $(p = 1)$, PolyDecayFast $(p = 2)$.

3. **Exponential Decay.** These matrices take the form

$$\boldsymbol{A} = \operatorname{diag}(\underbrace{1, \ldots, 1}_{R}, 10^{-q}, 10^{-2q}, \ldots, 10^{-(n-R)q}) \in \mathbb{F}^{n \times n}.$$

The parameter $q > 0$ controls the rate of exponential decay. We consider three examples: ExpDecaySlow $(q = 0.1)$, ExpDecayMed $(q = 0.25)$, ExpDecayFast $(q = 1)$.

**Application Examples.** We also consider **non-diagonal** matrices inspired by the SDP algorithm [44].

1. MaxCut: This is a **real-valued** psd matrix with dimension $n = 2\,000$, and its effective rank $R = 14$. We form approximations with rank $r \in \{1, 14\}$. The matrix is an approximate solution to the MAXCUT SDP [20] for the sparse graph G40 [10].

2. PhaseRetrieval: This is a psd matrix with dimension $n = 25\,921$. It has exact rank $250$, but its effective rank $R = 5$. We form approximations with rank $r \in \{1, 5\}$. The matrix is an approximate solution to a phase retrieval SDP; the data is drawn from our paper [44].

**Experimental Results.** Figures 5.1–5.2 display the performance of the three fixed-rank psd approximation methods for a subcollection of the input matrices. The vertical axis is the Schatten 1-norm

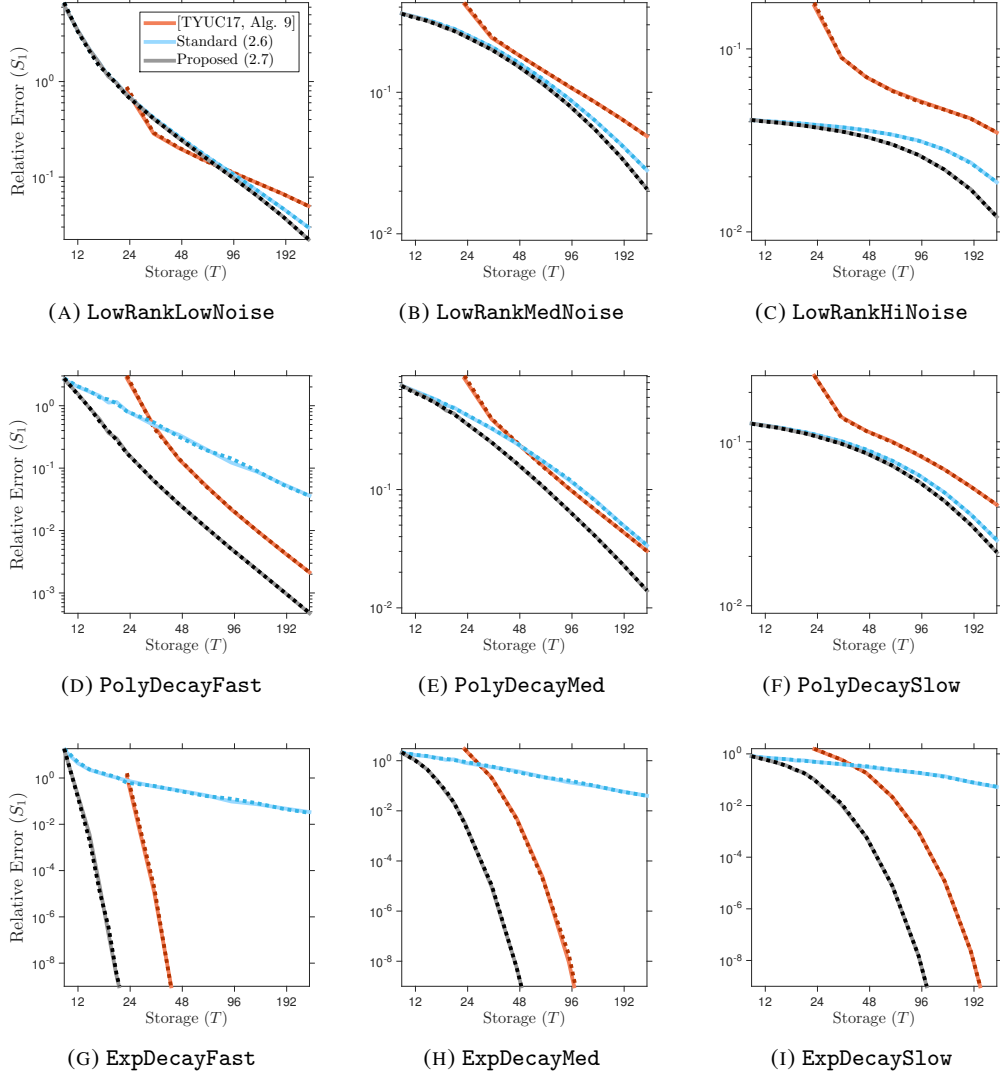

FIGURE 5.2: **Synthetic Examples with Effective Rank** $R = 10$, **Approximation Rank** $r = 10$, **Schatten** $1$-**Norm Error.** The data series show the performance of three algorithms for rank-$r$ psd approximation with $r = 10$. **Solid lines** are generated from the Gaussian sketch; **dashed lines** are from the SSFT sketch. Each panel displays the Schatten 1-norm relative error (5.1) as a function of storage cost $T$.

relative error (5.1). The variable $T$ on the horizontal axis is proportional to the storage required for the sketch only. For the Nyström-based approximations (2.6)–(2.7), we have the correspondence $T = k$. For the approximation [37, Alg. 9], we set $T = k + \ell$.

The experiments demonstrate that the proposed method (2.7) has a significant benefit over the alternatives for input matrices that admit a good low-rank approximation. It equals or improves on the competitors for almost all other examples and storage budgets. The supplement contains additional numerical results; these experiments only reinforce the message of Figures 5.1–5.2.

**Conclusions.** This paper makes the case for using the proposed fixed-rank psd approximation (2.7) in lieu of the alternatives (2.6) or [37, Alg. 9]. Theorem 4.1 shows that the proposed fixed-rank psd approximation (2.7) can attain any prescribed relative error, and Theorem 4.2 shows that it can exploit spectral decay. Furthermore, our numerical work demonstrates that the proposed approximation improves (almost) uniformly over the competitors for a range of examples. These results are timely because of the recent arrival of compelling applications, such as [44], for sketching psd matrices.

**Acknowledgments.** The authors wish to thank Mark Tygert and Alex Gittens for helpful feedback on preliminary versions of this work. JAT gratefully acknowledges partial support from ONR Award N00014-17-1-2146 and the Gordon & Betty Moore Foundation. VC and AY were supported in part by the European Commission under Grant ERC Future Proof, SNF 200021-146750, and SNF CRSII2-147633. MU was supported in part by DARPA Award FA8750-17-2-0101.

## Footnotes

[1] A signed permutation has exactly one nonzero entry in each row and column; the nonzero has modulus one.

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
