[Supplementary Material]

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

## A    Details of the Theoretical Analysis

This appendix contains a new theoretical analysis of the simple Nyström approximation (2.5) and the proposed fixed-rank Nyström approximation (2.7).

### A.1    Best Approximation in Schatten Norms

Let us introduce compact notation for the optimal rank-$r$ approximation error in the Schatten $p$-norm:

$$\sigma_{r+1}^{(p)}(\boldsymbol{M}) = \|\boldsymbol{M} - [\![\boldsymbol{M}]\!]_r\|_p = \left[\sum\nolimits_{i>r} \sigma_i(\boldsymbol{M})^p\right]^{1/p}. \tag{A.1}$$

Ordinary singular values correspond to the case $p = \infty$.

### A.2    Analysis of the Nyström Approximation

The first result gives a very accurate error bound for the basic Nyström approximation $\hat{\boldsymbol{A}}^{\mathrm{nys}}$ with respect to the Schatten 1-norm. This estimate is the key ingredient in the proof of Theorem 4.2.

**Theorem A.1** (Error in Nyström Approximation). *Assume $1 \le r \le k \le n$. Let $\boldsymbol{A} \in \mathbb{F}^{n \times n}$ be a psd matrix. Draw the test matrix $\boldsymbol{\Omega} \in \mathbb{F}^{n \times k}$ from the Gaussian or orthonormal distribution, and form the sketch $\boldsymbol{Y} = \boldsymbol{A}\boldsymbol{\Omega}$. Then the rank-$k$ Nyström approximation $\hat{\boldsymbol{A}}^{\mathrm{nys}}$ determined by (2.5) satisfies the error bound*

$$\mathbb{E}\,\|\boldsymbol{A} - \hat{\boldsymbol{A}}^{\mathrm{nys}}\|_1 \le \min_{\varrho < k-\alpha} \left[\left(1 + \frac{\varrho}{k - \varrho - \alpha}\right)\sigma_{\varrho+1}^{(1)}(\boldsymbol{A})\right]. \tag{A.2}$$

*The index $\varrho$ ranges over natural numbers. The quantities $\alpha(\mathbb{R}) = 1$ and $\alpha(\mathbb{C}) = 0$. The optimal rank-$\varrho$ Schatten 1-norm approximation error is defined in (A.1).*

To the best of our knowledge, Theorem A.1 is new. The proof appears below in App. A.3.

Let us situate Theorem A.1 with respect to the results in Gittens's work [18, 20]. Gittens develops error bounds for the Nyström approximation (2.5) that hold with high probability, rather than in expectation. He measures errors in the Schatten $p$-norm for $p = 1, 2, \infty$. He also obtains results for several types of test matrices, including isotropic models and a relative of the SSFT. In contrast to Theorem A.1, Gittens's bounds are more complicated, and the constants are much larger.

### A.3    Proof of Theorem A.1

We begin with the proof of Theorem A.1. Gittens [17, 18, 20] uses a related argument to obtain bounds on the *probability* that the Nyström approximation achieves a given error.

The first step is to write the Nyström approximation in terms of an orthogonal projector. This expression allows us to exploit the analysis from [23, 38].

**Proposition A.2** (Representation of Nyström Approximation). *Let $\boldsymbol{P}$ be the orthogonal projector onto* range$(\boldsymbol{A}^{1/2}\boldsymbol{\Omega})$:

$$\boldsymbol{P} = (\boldsymbol{A}^{1/2}\boldsymbol{\Omega})(\boldsymbol{\Omega}^*\boldsymbol{A}\boldsymbol{\Omega})^\dagger(\boldsymbol{A}^{1/2}\boldsymbol{\Omega})^*. \tag{A.3}$$

*Then the Nyström approximation (2.5) can be expressed as*

$$\hat{\boldsymbol{A}}^{\mathrm{nys}} = \boldsymbol{A}^{1/2}\boldsymbol{P}\boldsymbol{A}^{1/2} \tag{A.4}$$

*In particular, the Nyström approximation only depends on $\boldsymbol{\Omega}$ through* range$(\boldsymbol{\Omega})$.

We believe that Proposition A.2 first appeared explicitly in the work of Gittens [17].

*Proof.* This argument follows from a direct calculation:

$$\hat{A}^{\text{nys}} = A\Omega(\Omega^* A\Omega)^\dagger \Omega^* A$$
$$= A^{1/2}(A^{1/2}\Omega)\big[(A^{1/2}\Omega)^*(A^{1/2}\Omega)\big]^\dagger (A^{1/2}\Omega)^* A^{1/2}$$
$$= A^{1/2} P A^{1/2}.$$

To reach the last line, we identified the orthogonal projector (A.3). □

With Proposition A.2 at hand, the proof of Theorem A.1 is straightforward.

We may assume that $\Omega$ is a Gaussian matrix because the reconstruction $\hat{A}$ only depends on $\text{range}(\Omega)$. The range of a random orthonormal matrix has the same distribution as a Gaussian matrix up to a set of measure zero.

Let $P$ be the orthogonal projector (A.3). In view of the formula (A.4) for $\hat{A}^{\text{nys}}$, we have

$$A - \hat{A}^{\text{nys}} = A^{1/2}(I - P)A^{1/2}. \tag{A.5}$$

We can now express the Schatten 1-norm of the error in terms of the Schatten 2-norm:

$$\|A - \hat{A}^{\text{nys}}\|_1 = \|A^{1/2}(I-P)(I-P)A^{1/2}\|_1 = \|(I-P)A^{1/2}\|_2^2.$$

The first identity follows from (A.5) and the fact that the orthogonal projector $I - P$ is idempotent.

Fix a natural number $\varrho < k - \alpha$. We can use established results from the literature to control the expectation of the error. In particular, we invoke a slight generalization [38, Fact. 8.3] of a result [23, Thm. 10.5] of Halko et al. We arrive at the bound

$$\mathbb{E}\|(I - P)A^{1/2}\|_2^2 \leq \left(1 + \frac{\varrho}{k - \varrho - \alpha}\right) \sum_{i > \varrho} \sigma_i(A^{1/2})^2$$
$$= \left(1 + \frac{\varrho}{k - \varrho - \alpha}\right) \sum_{i > \varrho} \sigma_i(A) = \left(1 + \frac{\varrho}{k - \varrho - \alpha}\right) \sigma_{\varrho+1}^{(1)}(A).$$

Combine the last two displays and minimize over eligible $\varrho$ to complete the argument.

**Remark A.3** (Spectral-Norm Error)**.** When $\mathbb{F} = \mathbb{R}$, we can also obtain a spectral-norm error bound by combining this argument with another result [23, Thm. 10.6] of Halko et al.:

$$\mathbb{E}\sqrt{\|A - \hat{A}^{\text{nys}}\|} \leq \min_{\varrho < k-1} \left[\left(1 + \sqrt{\frac{\varrho}{k - \varrho - 1}}\right)\sqrt{\sigma_{\varrho+1}(A)} + \frac{\mathrm{e}\sqrt{k}}{k - \varrho}\sqrt{\sigma_{\varrho+1}^{(1)}(A)}\right].$$

It takes a surprising amount of additional work to obtain an accurate bound for the first moment of the error (instead of the 1/2 moment). We have chosen not to include this argument.

**Remark A.4** (High-Probability Bounds)**.** As noted by Gittens [18, 20], we can obtain high-probability error bounds in the real setting by combining the approach here with results [23, Thms. 10.7–10.8] from Halko et al. We omit the details.

**Remark A.5** (Other Test Matrices)**.** As noted by Gittens [18, 20], we can obtain results for other types of test matrices by replacing parts of the analysis that depend on Gaussian matrices. These changes result in bounds that are quantitatively and qualitatively worse. The numerical evidence suggests that many types of test matrices have the same empirical performance, so we omit this development.

## A.4 Theorem 4.2: Schatten 1-Norm Bound

Let us continue with the proof of the Schatten 1-norm bound (4.3) from Theorem 4.2. We require a basic result on rank-$r$ approximation adapted from [38, Prop. 7.1].

**Proposition A.6** (Fixed-Rank Projection)**.** *Let $A \in \mathbb{F}^{n \times n}$ and $\hat{A} \in \mathbb{F}^{n \times n}$ be arbitrary matrices. For each natural number $r$ and number $p \in [1, \infty]$,*

$$\|A - [\![\hat{A}]\!]_r\|_p \leq \sigma_{r+1}^{(p)}(A) + 2\|A - \hat{A}\|_p.$$

*Proof.* The argument follows from a short calculation based on the triangle inequality:

$$\|\boldsymbol{A} - [\![\hat{\boldsymbol{A}}]\!]_r\|_p \le \|\boldsymbol{A} - \hat{\boldsymbol{A}}\|_p + \|\hat{\boldsymbol{A}} - [\![\hat{\boldsymbol{A}}]\!]_r\|_p$$
$$\le \|\boldsymbol{A} - \hat{\boldsymbol{A}}\|_p + \|\hat{\boldsymbol{A}} - [\![\boldsymbol{A}]\!]_r\|_p$$
$$\le 2\|\boldsymbol{A} - \hat{\boldsymbol{A}}\|_p + \|\boldsymbol{A} - [\![\boldsymbol{A}]\!]_r\|_p.$$

In the second line, we have used the fact that $[\![\hat{\boldsymbol{A}}]\!]_r$ is a *best* rank-$r$ approximation of $\hat{\boldsymbol{A}}$. To complete the argument, we identify the last term (A.1) as the best rank-$r$ approximation error in the Schatten $p$-norm. $\square$

The bound (4.3) from Theorem 4.2 is now an immediate consequence of Theorem A.1 and Proposition A.6:

$$\mathbb{E}\,\|\boldsymbol{A} - \hat{\boldsymbol{A}}_r\|_1 \le \sigma_{r+1}^{(1)}(\boldsymbol{A}) + 2\,\mathbb{E}\,\|\boldsymbol{A} - \hat{\boldsymbol{A}}^{\mathrm{nys}}\|_1$$
$$\le \sigma_{r+1}^{(1)}(\boldsymbol{A}) + 2\,\min_{\varrho < k-\alpha}\left(1 + \frac{\varrho}{k - \varrho - \alpha}\right)\sigma_{\varrho+1}^{(1)}(\boldsymbol{A}).$$

We have used the definition (2.7) of our fixed-rank approximation: $\hat{\boldsymbol{A}}_r = [\![\hat{\boldsymbol{A}}^{\mathrm{nys}}]\!]_r$.

**Remark A.7** (Extensions). Given a bound on the error in the Nyström approximation (2.5) in the Schatten $p$-norm for any test matrix, this approach automatically yields an estimate for the associated fixed-rank psd approximation (2.7).

## A.5 Theorem 4.1: Schatten 1-Norm Bound

Next, we turn to the proof of the Schatten 1-norm bound (4.1) from Theorem 4.1. This argument is based on the same approach as Theorem A.1, but we require several additional ingredients from [18, 22, 23, 38].

As before, we may assume that $\boldsymbol{\Omega}$ is Gaussian. With probability one, the nonzero eigenvalues of $\hat{\boldsymbol{A}}^{\mathrm{nys}}$ are all distinct, so the best rank-$r$ approximation $\hat{\boldsymbol{A}}_r$ of $\hat{\boldsymbol{A}}^{\mathrm{nys}}$ is determined uniquely.

Let $\boldsymbol{P}$ be the orthogonal projector (A.3). According to (A.4), the Nyström approximation takes the form

$$\hat{\boldsymbol{A}}^{\mathrm{nys}} = \boldsymbol{A}^{1/2}\boldsymbol{P}\boldsymbol{A}^{1/2} = (\boldsymbol{A}^{1/2}\boldsymbol{P})(\boldsymbol{P}\boldsymbol{A}^{1/2}).$$

Let $\boldsymbol{Q}$ denote the orthogonal projector onto the range of $[\![\boldsymbol{P}\boldsymbol{A}^{1/2}]\!]_r$. Using the (truncated) SVD of the matrix $\boldsymbol{P}\boldsymbol{A}^{1/2}$, we can verify that the best rank-$r$ approximation $\hat{\boldsymbol{A}}_r$ of $\hat{\boldsymbol{A}}^{\mathrm{nys}}$ satisfies

$$\hat{\boldsymbol{A}}_r = [\![\boldsymbol{A}^{1/2}\boldsymbol{P}]\!]_r [\![\boldsymbol{P}\boldsymbol{A}^{1/2}]\!]_r = \boldsymbol{A}^{1/2}\boldsymbol{P}\boldsymbol{Q}\boldsymbol{P}\boldsymbol{A}^{1/2}$$

As in the proof of Theorem A.1, the Schatten 1-norm of the error satisfies

$$\|\boldsymbol{A} - \hat{\boldsymbol{A}}_r\|_1 = \|\boldsymbol{A} - \boldsymbol{A}^{1/2}\boldsymbol{P}\boldsymbol{Q}\boldsymbol{P}\boldsymbol{A}^{1/2}\|_1 = \|(\boldsymbol{I} - \boldsymbol{Q}\boldsymbol{P})\boldsymbol{A}^{1/2}\|_2^2.$$

Since $\mathrm{range}(\boldsymbol{Q}) \subset \mathrm{range}(\boldsymbol{P})$, we can rewrite this expression as

$$\|(\boldsymbol{I} - \boldsymbol{Q}\boldsymbol{P})\boldsymbol{A}^{1/2}\|_2^2 = \|(\boldsymbol{I} - \boldsymbol{P}\boldsymbol{Q}\boldsymbol{P})\boldsymbol{A}^{1/2}\|_2^2 = \|\boldsymbol{A}^{1/2} - \boldsymbol{P}[\![\boldsymbol{P}\boldsymbol{A}^{1/2}]\!]_r\|_2^2.$$

The last identity holds because $\boldsymbol{Q}\boldsymbol{P}\boldsymbol{A}^{1/2} = [\![\boldsymbol{P}\boldsymbol{A}^{1/2}]\!]_r$. A direct application of Gu's result [22, Thm. 3.5] yields

$$\|\boldsymbol{A}^{1/2} - \boldsymbol{P}[\![\boldsymbol{P}\boldsymbol{A}^{1/2}]\!]_r\|_2^2 \le \|(\boldsymbol{I} - \boldsymbol{P})[\![\boldsymbol{A}^{1/2}]\!]_r\|_2^2 + \sum_{i>r}\sigma_i(\boldsymbol{A}^{1/2})^2.$$

A direct application of the result [38, Prop. 9.2] shows that

$$\mathbb{E}\,\|(\boldsymbol{I} - \boldsymbol{P})[\![\boldsymbol{A}^{1/2}]\!]_r\|_2^2 = \frac{r}{k - r - \alpha}\sum_{i>r}\sigma_i(\boldsymbol{A}^{1/2})^2.$$

As before, we note that

$$\sum_{i>r}\sigma_i(\boldsymbol{A}^{1/2})^2 = \sigma_{r+1}^{(1)}(\boldsymbol{A}).$$

Taking an expectation and sequencing these displays, we arrive at

$$\mathbb{E}\,\|\boldsymbol{A} - \hat{\boldsymbol{A}}_r\|_1 \le \left(1 + \frac{r}{k - r - \alpha}\right)\sigma_{r+1}^{(1)}(\boldsymbol{A}).$$

This is the stated result (4.1).

### A.6 Theorems 4.1 and 4.2: Schatten $\infty$-Norm Bounds

Last, we develop the bounds (4.2) and (4.4) on the Schatten $\infty$-norm of the fixed-rank psd approximation (2.7) using a formal argument. We require the following result.

**Proposition A.8** (Reversed Eckart–Young). *Let $\boldsymbol{A}, \boldsymbol{B} \in \mathbb{F}^{n \times n}$ be matrices, and assume that* $\mathrm{rank}(\boldsymbol{B}) \leq r$. *Then*

$$\|\boldsymbol{A} - \boldsymbol{B}\|_\infty \leq \sigma_{r+1}(\boldsymbol{A}) + \left[ \|\boldsymbol{A} - \boldsymbol{B}\|_1 - \sigma_{r+1}^{(1)}(\boldsymbol{A}) \right].$$

The proof of Proposition A.8 follows from a minor change to [22, Thm. 3.4].

*Proof.* As a consequence of Weyl's inequalities [2, Thm. III.2.1], we have the bound

$$\sigma_{i+r}(\boldsymbol{A}) \leq \sigma_i(\boldsymbol{A} - \boldsymbol{B}) + \sigma_{r+1}(\boldsymbol{B}) = \sigma_i(\boldsymbol{A} - \boldsymbol{B}). \tag{A.6}$$

The last identity holds because $\mathrm{rank}(\boldsymbol{B}) \leq r$. It follows that

$$
\begin{aligned}
\|\boldsymbol{A} - \boldsymbol{B}\|_1 &= \sum_{i \geq 1} \sigma_i(\boldsymbol{A} - \boldsymbol{B}) \\
&= \sigma_1(\boldsymbol{A} - \boldsymbol{B}) + \sum_{i \geq 2} \sigma_i(\boldsymbol{A} - \boldsymbol{B}) \\
&\geq \|\boldsymbol{A} - \boldsymbol{B}\|_\infty + \sum_{i \geq 2} \sigma_{i+r}(\boldsymbol{A}) \\
&= \|\boldsymbol{A} - \boldsymbol{B}\|_\infty - \sigma_{r+1}(\boldsymbol{A}) + \sigma_{r+1}^{(1)}(\boldsymbol{A}).
\end{aligned}
$$

The first expression is the representation of the Schatten 1-norm in terms of singular values. The inequality is (A.6). Finally, we identify the best Schatten 1-norm error from (A.1). $\qquad\square$

To obtain the Schatten $\infty$-norm bound (4.2), we combine Proposition A.8 with the Schatten 1-norm bound (4.1):

$$
\begin{aligned}
\mathbb{E}\,\|\boldsymbol{A} - \hat{\boldsymbol{A}}_r\|_\infty &\leq \sigma_{r+1}(\boldsymbol{A}) + \left[ \mathbb{E}\,\|\boldsymbol{A} - \hat{\boldsymbol{A}}_r\|_1 - \sigma_{r+1}^{(1)}(\boldsymbol{A}) \right] \\
&\leq \sigma_{r+1}(\boldsymbol{A}) + \frac{r}{k - r - \alpha} \cdot \sigma_{r+1}^{(1)}(\boldsymbol{A}).
\end{aligned}
$$

Similarly, to obtain the Schatten $\infty$-norm bound (4.4), we combine Proposition A.8 with the Schatten 1-norm bound (4.3).

## B  Supplemental Numerics

This appendix documents additional numerical work. These experiments provide a more complete picture of the performance of the psd approximation methods.

- Figure B.1 contains a plot of the singular-value spectrum of each input matrix described in Sec. 5.

- Figures B.2–B.10 document the results of numerical experiments for the remaining parameter regimes outlined in Sec. 5. In particular, we consider all Schatten $p$-norm relative error measures for $p \in \{1, 2, \infty\}$ and all effective rank parameters $R \in \{5, 10, 20\}$ for the synthetic data. We omit the case $p = \infty, R = 20$ because the plots are uninformative.

- Figure B.11 gives evidence about the numerical challenges involved in implementing Nyström approximations, such as (2.7). Our implementation in Algorithm 3 is based on the Nyström approximation routine `eigenn` released by Tygert [39] to accompany the paper [28]. We compare with another implementation strategy described in the text of the same paper [28, Eqn. (13)]. It is surprising to discover very different levels of precision in two implementations designed by professional numerical analysts.

(A) Low-Rank + PSD Noise

(B) Polynomial Decay

(C) Exponential Decay

(D) `MaxCut` and `PhaseRetrieval`

FIGURE B.1: **Singular Values of Input Matrices.** These plots display the singular value spectra of the input matrices that appear in the experiments. See Sec. 5 for descriptions of the matrices.

(A) PhaseRetrieval ($r = 1$)

(B) PhaseRetrieval ($r = 5$)

(C) MaxCut ($r = 1$)

(D) MaxCut ($r = 14$)

FIGURE B.2: **Application Examples, Approximation Rank** $r$**, Schatten** 2**-Norm Error.** The data series are generated by three algorithms for rank-$r$ psd approximation. **Solid lines** are generated from the Gaussian sketch; **dashed lines** are from the SSFT sketch. Each panel displays the Schatten 2-norm relative error (5.1) as a function of storage cost $T$. See Sec. 5 for details.

FIGURE B.3: **Application Examples, Approximation Rank $r$, Schatten $\infty$-Norm Error.** The data series are generated by three algorithms for rank-$r$ psd approximation. **Solid lines** are generated from the Gaussian sketch; **dashed lines** are from the SSFT sketch. Each panel displays the Schatten $\infty$-norm relative error (5.1) as a function of storage cost $T$. See Sec. 5 for details.

FIGURE B.4: **Synthetic Examples with Effective Rank** $R = 5$**, Approximation Rank** $r = 10$**, Schatten** 1**-Norm Error.** The series are generated by three algorithms for rank-$r$ psd approximation with $r = 10$. **Solid lines** are generated from the Gaussian sketch; **dashed lines** are from the SSFT sketch. Each panel displays the Schatten 1-norm relative error (5.1) as a function of storage cost $T$. See Sec. 5 for details.

FIGURE B.5: **Synthetic Examples with Effective Rank** $R = 20$, **Approximation Rank** $r = 10$, **Schatten** 1**-Norm Error.** The series are generated by three algorithms for rank-$r$ psd approximation with $r = 10$. **Solid lines** are generated from the Gaussian sketch; **dashed lines** are from the SSFT sketch. Each panel displays the Schatten 1-norm relative error (5.1) as a function of storage cost $T$. See Sec. 5 for details.

FIGURE B.6: **Synthetic Examples with Effective Rank** $R = 5$, **Approximation Rank** $r = 10$, **Schatten** 2**-Norm Error.** The series are generated by three algorithms for rank-$r$ psd approximation with $r = 10$. **Solid lines** are generated from the Gaussian sketch; **dashed lines** are from the SSFT sketch. Each panel displays the Schatten 2-norm relative error (5.1) as a function of storage cost $T$. See Sec. 5 for details.

FIGURE B.7: **Synthetic Examples with Effective Rank** $R = 10$, **Approximation Rank** $r = 10$, **Schatten 2-Norm Error.** The series are generated by three algorithms for rank-$r$ psd approximation with $r = 10$. **Solid lines** are generated from the Gaussian sketch; **dashed lines** are from the SSFT sketch. Each panel displays the Schatten 2-norm relative error (5.1) as a function of storage cost $T$. See Sec. 5 for details.

FIGURE B.8: **Synthetic Examples with Effective Rank** $R = 20$**, Approximation Rank** $r = 10$**, Schatten** 2**-Norm Error.** The series are generated by three algorithms for rank-$r$ psd approximation with $r = 10$. **Solid lines** are generated from the Gaussian sketch; **dashed lines** are from the SSFT sketch. Each panel displays the Schatten 2-norm relative error (5.1) as a function of storage cost $T$. See Sec. 5 for details.

FIGURE B.9: **Synthetic Examples with Effective Rank** $R = 5$, **Approximation Rank** $r = 10$, **Schatten $\infty$-Norm Error.** The series are generated by three algorithms for rank-$r$ psd approximation with $r = 10$. **Solid lines** are generated from the Gaussian sketch; **dashed lines** are from the SSFT sketch. Each panel displays the Schatten $\infty$-norm relative error (5.1) as a function of storage cost $T$. See Sec. 5 for details.

FIGURE B.10: **Synthetic Examples with Effective Rank** $R = 10$**, Approximation Rank** $r = 10$**,
Schatten** $\infty$**-Norm Error.** The series are generated by three algorithms for rank-$r$ psd approximation
with $r = 10$. **Solid lines** are generated from the Gaussian sketch; **dashed lines** are from the SSFT
sketch. Each panel displays the Schatten $\infty$-norm relative error (5.1) as a function of storage cost $T$.
See Sec. 5 for details.

(A) `ExpDecayFast`, $R = 5$        (B) `ExpDecayMed`, $R = 5$        (C) `ExpDecaySlow`, $R = 5$

(D) `ExpDecayFast`, $R = 10$     (E) `ExpDecayMed`, $R = 10$     (F) `ExpDecaySlow`, $R = 10$

(G) `ExpDecayFast`, $R = 20$     (H) `ExpDecayMed`, $R = 20$     (I) `ExpDecaySlow`, $R = 20$

FIGURE B.11: **Bad Numerics, Approximation Rank** $r = 10$**, Schatten** $1$**-Norm Error.** The series are generated by two implementations of the fixed-rank psd approximation (2.7). We compare Algorithm 3 with another approach [LLS+17] proposed in [28, Eqn. (13)]. **Solid lines** are generated from the Gaussian sketch; **dashed lines** are from the SSFT sketch. Each panel displays the Schatten 1-norm relative error (5.1) as a function of storage cost $T$. See App. B for details.

## Footnotes

[1] A signed permutation has exactly one nonzero entry in each row and column; the nonzero has modulus one.