[Reviews · NeurIPS 2017]

Reviewer 1



The paper considers the problem of finding a rank k approximation for a PSD matrix A in the streaming model. The PSD matrix is given as a sequence of updates A = A + H where H is a symmetric matrix. The paper considers a variant of a popular Nystrom method for this task. The algorithm stores the product Y = A*Omega where Omega can be a random Gaussian matrix. The paper proposes a different way to recover an approximation to A from Y from previous works. Technically, the paper is very similar to previous works by Gittens and Halko et al. The proofs are fairly simple (a bonus in my opinion) as the sketching matrix can be assumed to be Gaussian and the distribution of Y = A*Omega (Omega is the Gaussian matrix) is well understood. The downside is that the proofs are probably hard to generalize to other types of sketching matrices, such as SSFT mentioned in the paper. The experiments are promising as the proposed method seems to outperform previous algorithms. The new algorithm is a small modification of previous algorithms and uses the same sketch: the difference is only in the routine to obtain the approximation from the sketch. Thus, it can be plugged in and replace previous algorithms without much modification to the rest of the system.

Reviewer 2



The authors address the problem of fixed-rank approximation of a positive semidefinite (psd) matrix from streaming data. The authors propose a simple method based on the Nystrom approximation, in which a sketch is updated as streaming data comes in, and the rank-r approximation is obtained as the best rank-r approximation to the full Nystrom approximation. For Gaussian or uniformly distributed (Haar) orthogonal sketch matrices, the authors prove a standard upper bound on the Schatten-1 (aka Nuclear) norm of the approximation error. Detailed empirical evidence are discussed, which indicate that the proposed method improves on previously proposed methods for fixed-rank approximation of psd matrices. The paper is very well written and easy to follow. The result appears to be novel. The proposed method admits a theoretical analysis and seems to perform well in practice. Implementation details are discussed. This paper whould be a welcome addition to the steaming PCA literature and is a clear accept in my opinion. Minor comment: - The "experimental results" section does not mention which test matrix ensemble was used, and whether the true underlying rank r was made available to the algorithms. - Some readers will be more familiar with the term "nuclear norm" than with the term "Schatten-1 norm" - It would be nice to discuss the increase in approximation error due to streaming, compared with a classical sketch algorithm that is given all the data in advance.

Reviewer 3



This paper describes a different way of estimating a low rank PSD matrix from a sketched PSD matrix. From what I can tell, the main difference is that, in the usual case, the sketched test matrices (Y) may be far from orthogonal, and low-ranking the middle matrix itself may produce bad results. Therefore the paper proposes an efficient way of low-ranking the entire Nystrom-estimated matrix, and shows much better results. I believe this paper is very well written, with context well described and competing methods evenly mentioned. Suggestion: lines 138-143, which essentially just restates alg. 3, is not very enlightening. specifically I don't understand the need for the Cholesky factorization, since it does nothing to make things thinner. We can just as easily take the SVD of Y_nu*B, which is equally thin, and if the shift is large enough then everything should be stable, without the need of the extra factorization step. I suggest replacing this segment with an explanation rather than restatement of the alg.